# Bidirectional Associations between Parental Non-Responsive Feeding Practices and Child Eating Behaviors: A Systematic Review and Meta-Analysis of Longitudinal Prospective Studies

**DOI:** 10.3390/nu14091896

**Published:** 2022-04-30

**Authors:** Jian Wang, Bingqian Zhu, Ruxing Wu, Yan-Shing Chang, Yang Cao, Daqiao Zhu

**Affiliations:** 1School of Nursing, Shanghai Jiao Tong University, Shanghai 200025, China; jian.3.wang@kcl.ac.uk (J.W.); zhubq@shsmu.edu.cn (B.Z.); wu_ruxing@163.com (R.W.); 2Florence Nightingale Faculty of Nursing, Midwifery and Palliative Care, King’s College London, London SE1 8WA, UK; yan-shing.chang@kcl.ac.uk; 3Clinical Epidemiology and Biostatistics, School of Medical Sciences, Örebro University, 70182 Örebro, Sweden

**Keywords:** children, parents, feeding practices, eating behaviors, prospective study, meta-analysis, systematic review

## Abstract

Background: Parental non-responsive feeding practices and child eating behaviors both play significant roles in childhood obesity. However, their longitudinal relationships are less clear. This systematic review aimed to examine their bidirectional associations. Methods: A systematic search of five databases was conducted from inception to February 2022. Data synthesis was performed using a semi-quantitative and quantitative approach. Results: A total of 14 studies with 15348 respondents were included. A total of 94 longitudinal effects from 14 studies of parental non-responsive feeding practices on child eating behaviors were investigated, and 19 statistically significant effects were discovered. Seventy-seven longitudinal effects from nine studies of child eating behaviors on parental feeding practices were examined, with fifteen being statistically significant. The pooled results of meta-analysis showed five statistically significant associations: parental restrictive feeding positively predicted child enjoyment of food (β = 0.044; 95% CI: 0.004, 0.085); use of food as a reward positively predicted child emotional eating (β = 0.09; 95% CI: 0.04, 0.15); child food responsiveness positively predicted restrictive feeding (β = 0.04; 95% CI: 0.02, 0.06); use food as a reward (β = 0.06; 95% CI: 0.03, 0.10). In addition, the pooled effects showed that child satiety responsiveness negatively predicted restrictive feeding (β = −0.05; 95% CI: −0.08, −0.01). Conclusions: The bidirectional relationships between parental non-responsive feeding practices and child eating behaviors are inconsistent and a few showed statistical significance. Theory-driven longitudinal studies using validated instruments and controlling for potential confounders are needed to unveil their relationships and provide evidence for obesity prevention interventions.

## 1. Introduction

Childhood overweight and obesity is a worldwide health issue that affected 39 million children under the age of 5 in 2020 [1]. In China, 6.8% of children under the age of 6 were overweight and 3.6% were obese in 2020, accounting for the largest child population with obesity in the world [2]. Given the myriad of associated short-and long-term health consequences, e.g., hyperlipidemia, hypertension, and heart disease [3,4,5], early prevention of obesity has been a major focus for both research and policy [6].

Obesity in young children is a multi-faceted problem (e.g., genetic predisposition, energy consumption) [7,8], of which parental feeding practices and children’s eating behaviors have shown to play significant roles [9,10]. Feeding practices refer to specific practices or strategies that parents employ to manage what, when, and how much their children eat and shape their children’s eating patterns [11,12,13]. There are two types of parental feeding practices: non-responsive and responsive feeding [14,15,16]. Non-responsive feeding practices (also known as coercive control), such as applying pressure to eat, restricting food, and using food as a reward [13], have raised great concerns due to their close links to children’s obesity [9,17,18,19,20]. Positive relationships between non-responsive feeding practices and child weight status have been consistently reported [9,20]. Based on a recent meta-analysis of 51 studies with 17,431 parent-child dyads, the higher use of controlling feeding practices was associated with a greater risk of obesity [9]. Unlike adults, young children cannot choose the environment in which they live or the food they eat. Parental feeding practices are thus critical in shaping the eating behaviors of young children, both of which are important in addressing childhood overweight/obesity [21,22]. However, studies on the associations between parental feeding practices, particularly non-responsive feeding practices, and child eating behaviors have shown inconsistent results.

Some cross-sectional studies have reported the relationships between parental non-responsive feeding practices and child eating behaviors [23,24,25,26,27,28]. A population-based study (*n* = 4987) in the Netherlands showed that parental restrictions of food and pressure to eat were both positively correlated with four-year-old children’s food responsiveness, emotional overeating, and satiety responsiveness [28]. Similarly, a cross-sectional study (*n* = 977) in Australia indicated that parental higher overt restriction was associated with more preschool children’s food responsiveness [26]. Although prior studies cannot determine the causality between feeding practices and eating behaviors, theories of developmental psychology, such as Ecological System Theory, have suggested that the relationship between parental non-responsive feeding practices and child eating behaviors may be bidirectional [29]. Ventura and Birch (2008) [12] also proposed a model, suggesting that parenting (e.g., parenting styles and child-feeding styles) and child eating (e.g., eating styles, food preferences, and food intake) are interrelated.

Recent longitudinal studies provided empirical support for bidirectional relationships [14,30,31]. For instance, Jansen et al. [31] found that high levels of mothers’ pressure to eat at age five of the child were associated with higher food fussiness at age six. Conversely, child food fussiness at age one and a half and three was positively associated with food pressure at age four. However, other studies did not find such a bidirectional relationship. For example, a Norway cohort study (*n* = 797) showed that greater parental use of food as a reward positively predicted child emotional eating and food responsiveness with a 2-year follow-up, while child eating behaviors did not predict parental feeding practices (i.e., parent-driven association) [32]. Similarly, a study conducted in the US (*n* = 229) showed that mothers’ restriction of food amount at 21 months of the child was a negative predictor for child eating in the absence of hunger (EAH) at 27 months, but EAH did not prospectively predict maternal food restriction [33]. Lumeng et al. [34] did not find bidirectional associations between pressure feeding and food fussiness.

Overall, current findings of the bidirectional relationships between parental non-responsive feeding practices and child eating behaviors have been inconsistent. Given the increasing prevalence of overweight and obesity among young children and progressive evidence linking parental non-responsive feeding practices to child eating behaviors, the aim of this systematic review was to identify and summarize the relationships of interest reported previously. Findings from this review may enhance our understanding of their bidirectional correlations, provide recommendations for future research, and guide future interventions for the prevention of childhood obesity.

## 2. Materials and Methods

### 2.1. Design

The systematic review and meta-analysis were guided by the Preferred Reporting Items for Systematic Reviews and Meta-Analysis (PRISMA) guidelines [35] and the Meta-analysis of Observational Studies in Epidemiology (MOOSE) [36]. This review was registered in PROSPERO (registration number: CRD42020183287).

### 2.2. Data Sources and Search Strategy

A systematic search (from inception to February 2022) was carried out on PubMed, Embase, PsycINFO, ScienceDirect, and Cochrane Library. We also searched for grey literature on Virtual Health Library (http://bvsalud.org/en/, accessed on 12 February 2022), NARCIS (https://www.narcis.nl/, accessed on 12 February 2022), Grey literature report (http://greylit.org/, accessed on 12 February 2022), and Open grey EU (http://opengrey.eu/, accessed on 12 February 2022). The search was limited to publications in English. Free text and the Medical Subject Headings (MeSH) terms were used for the search. The following terms were used with Boolean operators: ‘child’ AND ‘feeding’ AND (‘eating’ OR ‘diet’ OR ‘food intake’) AND (‘cohort’ OR ‘longitudinal’ OR ‘interaction’ OR ‘prospective’). A manual search of the bibliography of included studies was performed to identify additional studies.

### 2.3. Inclusion and Exclusion Criteria

All studies examining the longitudinal relationships between parental non-responsive feeding practices and child eating behaviors were included if they met the following criteria:(1)Both parental feeding practices and child eating behaviors were reported.(2)Children aged 1–6 years at baseline (evidence showed that energy intake regulation and establishment of healthy eating are effective when the children are 1–6 years [37,38]).

Studies were excluded if they possessed the following features:
(1)Were reviews, editorials, commentaries, letters, or methodological papers;(2)Were non-English papers;(3)Did not report the bidirectional relationships between caregivers’ non-responsive feeding practices and children’s eating behaviors;(4)Focused on children with diseases that might influence their eating;(5)Included primary caregivers who were not the child’s parents;(6)Used observation for the assessment of feeding practices or eating behaviors. Observational assessment may be subject to the constraints of specific occasions and, thus, cannot represent the meaning of behavior completely, while self-reports tend to refer to general practices [39,40]. Additionally, self-reports of feeding showed greater stability over time than observational measures [41]. This large discrepancy between observed and self-reported information may result in methodological inconsistency across the studies.

### 2.4. Study Screening and Data Extraction

The PRISMA flow chart was followed during the screening stage [35]. One investigator (J.W.) screened the title and abstract for initial inclusion. Full texts were reviewed independently by the two investigators (J.W. and R.W.) for further screening. Data extracted from each study included author, year of publication, and country. We also extracted participant- and study-related characteristics, such as parental age, children’s age, study duration, sampling method, sample size, response rate, variables of interest and their measures, and main findings. These data were tabulated into tables developed by the review team. For any disagreements that occurred during the screening and data extraction stages between the two investigators, a third reviewer was consulted (D.Z.).

### 2.5. Primary and Secondary Outcomes

Based on the conceptual analysis of food parenting practices [42,43,44], non-responsive feeding practices were classified into four categories including restriction, pressure to eat, emotional feeding, and use food as a reward:(1)Restriction means that the caregivers enforce strict limitations on the child’s access to food or opportunities to consume specific food [45]. Typically, restrictive feeding practices are used to control child’s intake of unhealthy food [30,46,47,48]. Restriction for weight is another common restrictive feeding [46].(2)Pressure to eat means that caregivers insist, demand, or physically struggle with the child in order to get the child to eat more food [45,46,47,48].(3)Emotional feeding represents caregivers using food as a method to manage or calm the child when he/she is upset, fussy, angry, hurt, or bored [45,49]. For example, food to soothe, food for boredom, and food for stress.(4)Use food as a reward, also called instrumental feeding, involves using threats and bribes or pushing children to eat more [45,50]. Based on the conceptual model proposed by Vaughn [45], using food as a reward was classified into reward for eating, reward for behavior, and nonfood-based incentives to eat.
Reward for behavior means that caregivers threaten to take something away for misbehavior or promise/offer something to the child in return for desired behavior [45,50].Reward for eating means that threats and bribes can be used to manage child’s behavior for the purposes of general obedience or behaviors specific to eating [45,50].Nonfood incentives to eat means that threats and bribes around child eating behaviors (e.g., eating disliked food) may be nonfood incentives (e.g., stickers) [45].

Child eating behaviors were categorized into five types, which are commonly used in previous studies [32,51,52,53]:(1)Food fussiness refers to being highly selective about the range of food that are accepted [51]. It is assessed by using a range of ad hoc measures, which are often based on lists of food that might be accepted or rejected [51,54].(2)Satiety responsiveness refers to the amount of food the child eats in a meal [54]. This is usually measured behaviorally by seeing whether food intake is reduced to compensate for a prior snack [51].(3)Food responsiveness refers to the desire to eat food when they see or smell food or are supplied with food [54]. It is assessed behaviorally on the basis of the amount of good-tasting versus less-good-tasting food consumed in normal conditions [51]. The items of food responsiveness are on self-reported desire for food following exposure to attractive food cues [51,55].(4)Emotional eating refers to eating in response to emotions (e.g., happiness, anger, worrying, and depression) [54]. It usually refers to eating more food during negative emotional states, although recent work has begun to distinguish emotional overeating from emotional undereating [51,56].(5)Enjoyment of food refers to the extent of enjoying all kinds of food and desire to eat [51,54]. The opposite characteristic, lack of interest in food, emerges as a common problem in the literature [51].

### 2.6. Quality Appraisal

The Joanna Briggs Institute (JBI) Critical Appraisal Checklist for Cohort Studies was used for quality appraisal [57]. This tool assesses the methodological quality of a study and to determines the extent to which a study has addressed the possibility of bias in its design, conduct and analysis. Two independent reviewers (J.W. and R.W.) performed the assessment, checking for possible sources of bias, attrition, and validity of survey instruments.

### 2.7. Data Analyses

Statistical parameters representing the associations between parental non-responsive feeding practices and child eating behaviors were extracted by regression coefficient (β) with their 95% confidence interval (CI) or standard error (adjusted for covariates). We contacted the authors if their studies did not report these statistics. The studies eventually without available standard errors or 95% CIs of regression coefficients were excluded from the meta-analysis. We thus used a semi-quantitative approach for data synthesis of all included articles to describe their associations across the time points, as adopted by recent reviews [58,59].

The studies with necessary information were included in the meta-analysis. For associations evaluated multiple time points in one study, they were first synthesized within the study, and the summarized data were used for meta-analysis. In examining the longitudinal associations between specific feeding practices and child eating behaviors, the pooled regression coefficients with corresponding 95% CI across studies were calculated and presented using forest plots. Random-effects models were used when heterogeneity among the studies was presented [60]. The degree of heterogeneity in individual effect was assessed by using *I*^2^ statistic, in which *I*^2^ > 30% was considered to indicate moderate heterogeneity and *I*^2^ > 50% was considered to indicate substantial heterogeneity. A *p* value < 0.05 from the noncentral chi-squared test for heterogeneity was considered to indicate statistically significant heterogeneity [61]. The publication bias was evaluated using the Egger’s test. Because no specific associations were evaluated in more than three studies, funnel plots were not provided. For the pooled results, a 95% CI that did not include 0 was considered statistically significant. All analyses were performed in Stata 17.0 (StataCorp, College Station, TX, USA).

## 3. Results

### 3.1. Search Results

A total of 2483 articles were identified. The removal of duplicates resulted in 1982 articles for initial screening, and 249 articles were retrieved. After screening the full texts of 249 articles, 14 prospective studies were included. The PRISMA flow chart is shown in Figure 1.

### 3.2. Quality Appraisal

Eleven studies were rated high quality. Three studies were rated moderate quality (see Appendix A).

### 3.3. Characteristics of the Studies

Characteristics of the studies are shown in Table 1. The studies were conducted between 2003 and 2021 in the US (*n* = 3) [34,62,63], Netherlands (*n* = 2) [31,53], Norway (*n* = 2) [32,64], Portugal (*n* = 1) [65], and Australia (*n* = 6) [14,30,40,52,66,67]. The total number of participants was 15348, with individual study sample size ranging from 72 [40] to 4845 [31]. The caregivers were typically mothers (*n* = 9847; 10 studies) and parents (*n* = 5501; four studies). Nine studies used voluntary (response) sampling and the response rate ranged from 59.83% [14,66] to 100% [62,63,67].

Seven studies collected data at two time points [30,32,40,52,53,65,67], of which four had one-year follow-up [30,40,52,67], one had two-year [32], one had three-year [65], and one had five-years [53]. Five studies collected data at three time points [14,34,62,64,66]. The frequency of the follow-up included every six months [34], every one year [62], every two years [64], and 1.7 years at the first time and 1.3 years at the second time [14,66]. Two studies assessed child eating behaviors at four time points [31,63].

### 3.4. Measurements for Parental Feeding Practices

Tools used for the assessment of parental feeding practices are shown in Table 2. The most common tool (*n* = 7) [31,40,52,53,62,65,67] was the Child Feeding Questionnaire (CFQ) [48], which was intended for use by parents of children aged 2–11 years. Others included the Parent Feeding Style Questionnaire (PFSQ) [32] (*n* = 3) [32,63,64], the Feeding Practices and Structure Questionnaire (FPSQ) [50] (*n* = 2) [14,66], the Infant Feeding Styles Questionnaire (IFSQ) [68] (*n* = 1) [34], and the Eating Disorder Examination Questionnaire (EDE-Q) [69] (*n* = 1) [64]. In addition, Rodgers et al. [30] conducted principal component analyses (PCA) to explore the components of maternal feeding practices, which included five main measures assessing feeding practices. 

### 3.5. Measurements for Child Eating Behaviors

Tools used for the assessment of child eating behaviors are also shown in Table 2. Eleven studies used the Children’s Eating Behavior Questionnaire (CEBQ) [51] to assess domains of child eating, including food responsiveness [14,32,52,53,63,64,65], satiety responsiveness [14,32,63], food fussiness [31,34,40,52,63,65,66,67], emotional eating [32], and enjoyment of food [32,40,52,63,67]. One study [53] used CEBQ [51] to assess a child’s food responsiveness, emotional overeating, and satiety responsiveness and the Stanford Feeding Questionnaire (SFQ) [70] to assess food fussiness. Other instruments included the Child Behavior Checklist [71], Foods I like and Dislike (FILAD) [62], and the Dutch Eating Behavior Questionnaire (DEBQ-P) [72].

### 3.6. Longitudinal Associations between Parental Non-Responsive Feeding Practices and Child Eating Behaviors

The longitudinal associations are shown in Appendix A. All studies examined the longitudinal effects of parental non-responsive feeding practices on child eating behaviors (F→E). The outcome variables (FàE) included food fussiness (*n* = 10) [31,34,40,52,53,62,63,65,66,67], food responsiveness (*n* = 7) [14,32,52,53,63,64,65], satiety responsiveness (*n* = 4) [14,32,53,63], emotional eating (*n* = 3) [30,32,53], and enjoyment of food (*n* = 5) [32,40,52,63,67]. Ten studies also tested their reverse relationships (E→F). The outcome variables (E→F) related to parental non-responsive feeding practices were restriction (*n* = 5) [14,32,63,65,66], pressure to eat (*n* = 3) [31,34,65], use food as a reward (*n* = 6) [14,30,32,53,63,66], and emotional feeding (*n* = 2) [30,63].

#### 3.6.1. Longitudinal Effects of Parental Non-Responsive Feeding Practices on Child Eating Behaviors (F→E)

Table 3 summarizes the longitudinal effects of parental non-responsive feeding practices on child eating behaviors. A total of 94 longitudinal effects (F→E) were investigated in 14 studies, with 19 significant effects discovered. These studies focused on examining the longitudinal effects of parental restriction of food (*n* = 10) [14,32,40,52,62,63,64,65,66,67], pressure to eat *(n* = 7) [31,34,40,52,62,65,67], and use of food as a reward (*n* = 6) [14,32,53,63,64,66] on child eating behaviors. The longitudinal influence of parental pressure to eat on children’s food fussiness was most frequently examined (*n* = 7) [31,34,40,52,62,65,67], revealing that parental higher pressure to eat significantly predicted increased likelihood of children’s food fussiness [31,65].

Specifically, 10 studies [14,32,40,52,62,63,64,65,66,67] examined 33 longitudinal effects of parental restrictive feeding on child eating behaviors. Only one study found a negative effect of parental restrictive feeding on children’s enjoyment of food [40]. Seven studies [31,34,40,52,62,65,67] tested 15 longitudinal effects of parental pressure to eat on child eating behaviors, with five of them having statistical significance. The results showed that parental pressure to eat positively predicted child food fussiness [31,65] and food responsiveness [65] while it negatively predicted enjoyment of food [40,52]. Six studies [14,32,53,63,64,66] (33 longitudinal effects were tested) reported significant effects of parental using food as a reward on child eating behaviors. Parental use of food as a reward positively predicted children’s food fussiness [53,66], food responsiveness [14,32,63], and emotional eating [32,53]. Finally, two studies [30,63] tested 13 longitudinal effects of parental emotional feeding on child eating behaviors and showed that parental emotional feeding had a positive association with food responsiveness [63] and emotional eating [30]. However, evidence for the link between parental emotional feeding and satiety responsiveness was contradictory [63]. That is, higher parental emotional feeding at age 3.3 years of the child predicted higher child satiety response at 4.3 years, but higher parental emotional feeding at 4.3 years was associated with lower child satiety responsiveness at 5.3 years.

Figure 2, Figure 3 and Figure 4 presents effects with 95% CIs for assessing the associations of parental non-feeding practices on child eating behaviors. For associations that were evaluated in two or more studies, two associations are statistically significant where parental higher restriction of food predicted higher children’s enjoyment of food (β = 0.044; 95% CI: 0.004, 0.085) and parental higher use of food as a reward predicted higher children’s emotional eating (β = 0.09; 95% CI: 0.04, 0.15).

#### 3.6.2. Longitudinal Effects of Child Eating Behaviors on Parental Non-Responsive Feeding Practices (E→F)

Table 4 summarizes the longitudinal effects of child eating behaviors on parental non-responsive feeding practices. Nine studies tested 77 longitudinal effects (E→F) and found 15 significant associations. These studies focused on examining the longitudinal effects of children’s food fussiness (*n* = 6) [31,34,53,63,65,66], food responsiveness (*n* = 5) [14,32,53,63,65], and satiety responsiveness (*n* = 4) [14,32,53,63] on parental non-responsive feeding practices. The longitudinal influence of children’s food responsiveness on parental use of food as a reward was commonly examined (*n* = 4) [14,32,53,63] and two studies reported that children’s higher level of food responsiveness predicted higher parental use of food as a reward [53,63].

Six studies [31,34,53,63,65,66] examined 24 longitudinal influences of children’s food fussiness on parental non-responsive feeding practices. Four studies showed that children’s food fussiness positively predicted parental pressure to eat [31,65] and use of food as a reward [66] while it negatively predicted parental emotional feeding [63]. Five studies [14,32,53,63,65] examined 20 longitudinal effects of children’s food responsiveness on parental non-responsive feeding practices, with five statistically significant relationships. Specifically, children’s food responsiveness positively predicted parental restrictive feeding [65] and use of food as a reward [53,63] and negatively predicted their pressure to eat [65]. Four studies [14,32,53,63] tested 18 longitudinal effects of children’s satiety responsiveness on parental non-responsive feeding practices, with two significant relationships, showing that children’s satiety responsiveness positively predicted parental restrictive feeding [14] and negatively predicted parental use of food as a reward [63]. Three studies [30,32,53] reported longitudinal effects of children’s emotional eating on parental non-responsive feeding practices. Four effects were tested, with two of them significant: children’s emotional eating positively predicted parental use of food as a reward [53] and emotional feeding [30]. Finally, two longitudinal studies [32,63] examined 11 longitudinal effects of children’s enjoyment of food on parental non-responsive feeding practices, with one significant finding in which children’s enjoyment of food positively predicted parental use of food as a reward [63].

Figure 5, Figure 6, Figure 7 and Figure 8 presents effects with 95% CIs for assessing the associations of child eating behaviors with parental non-responsive feeding practices. For associations that were evaluated in two or more studies, child food responsiveness positively predicted parental restrictive feeding (β = 0.04; 95% CI: 0.02, 0.06) and used food as a reward (β = 0.06; 95% CI: 0.03, 0.10). Child satiety responsiveness was also associated with decreased likelihood of parental restrictive feeding (β = −0.05; 95% CI: −0.08, −0.01).

However, all five synthesized results were only based on available data from no more than three studies and heterogeneity between studies was also considerable (*I*^2^ between 29.7% and 55.7%).

## 4. Discussion

The aim of this systematic review and meta-analysis was to summarize available evidence about the bidirectional relationships between parental non-responsive feeding practices and child eating behaviors. The fourteen included studies examined the longitudinal influence of parental non-responsive feeding practices on child eating behaviors, and nine of them also tested the longitudinal influence of child eating behaviors on parental non-responsive feeding practices. Overall, the bidirectional associations between parental non-responsive feeding practices and child eating behaviors are mixed. Only weak longitudinal correlations were found between some specific feeding practices and eating behaviors.

A total of 14 studies tested 94 longitudinal effects of parental non-responsive feeding practices on child eating behaviors and reported 19 significant associations. The results suggested that parental non-responsive feeding practices might result in child unhealthy eating behaviors. In particular, the finding of meta-analysis indicated that higher levels of parental use of food as a reward predicted increased likelihood of children’s emotional eating. Parents tend to use food as a symbol for their love towards their children or as an educational and emotional tool for shaping their children’s behaviors [73]. This behavior may commonly make children feel delighted, which may affect their normal attitudes and emotions toward food, especially their favorite food. As a result, if children become emotional (e.g., sadness), they may seek to comfort and soothe with food [54]. We also found that higher parental restrictive feeding might predict children’s more interest in food. Higher parental use of coercive control feeding such as restriction of food (e.g., sugar-sweetened intake) might inhibit child self-regulatory eating, resulting in children’s more interest in the restricted food [65].

Nine studies examined the prospective influence of child eating behaviors on parental feeding practices. Seventy-seven longitudinal effects were tested, and significant associations were detected in fifteen tests. It might be concluded that children’s unhealthy eating behaviors (e.g., food responsiveness, and satiety responsiveness) was positively associated with parental non-responsive feeding practices. The results of meta-analysis showed that children’s food responsiveness positively predicted parental use of food as a reward and restrictive feeding. The explanation for our finding is that, if a child likes to eat or eats more, food is an obvious reward for parents to give [63] because they can use this method to shape children’s behaviors [45]. It is also noting that children’s overeating may make parents be concerned about the child eating too much and being overweight or obese in the future, which may eventually result in their higher restrictions on food [74,75]. We also found that child satiety responsiveness negatively predicted parental restrictive feeding. Children’s eating too little may reinforce parental concerns about child weight and nutrition [21], which may decrease the likelihood of adopting restrictive feeding (e.g., restriction of food amount and variety) [14,76].

Theoretically, feeding and eating behaviors interact with each other; however, current findings from longitudinal studies do not provide strong evidence. There may be several reasons. First, age may play a role in the non-significant findings. For example, Bauer et al. [33] found that neither the restriction of food quantity nor food quality increased the risk for children’s EAH in toddlers below 3 years old. This could be due to age-related cognitive differences in children [77]. Compared to younger children, older children may be more aware that their intake is being limited by their parents [76] and they may respond to this feeding, such as EAH [78]. Second, the time of follow-up may be an important factor. For example, Steinsbekk et al. [32] found that child eating behaviors did not predict parental feeding practices two years later, whereas Rodgers et al. [30] reported that some child effects emerged one year later. Longer follow-up may miss critical periods of changes in their relationships. Third, the individual study may be under-powered to detect the significant association. A randomized controlled trial found that mothers’ feeding practices explained the variance in changes of child eating behaviors, but the effect size was small [79]. Given that many factors are related to non-responsive feeding practices and eating behaviors [73,80,81,82], a larger sample size is required in observational studies.

We also found inconsistent findings about the longitudinal associations between specific feeding practices and eating behaviors. The reasons are as follows. Child sex may be a contributing factor for the inconsistency [81,83]. Mothers’ feeding control may differ depending on the sex of their children [84]. It is possible that mothers with girls may be conscious of higher societal body shape expectations placed on females and, therefore, view their engagement in their daughters eating as protective against unhealthy eating and weight [85,86]. Eventually, girls may be more likely to overeat in response to the presence of palatable foods [78]. Participants varied across studies, which may also explain the inconsistency. Mothers were the respondents in ten studies and parents were the respondents in the remaining four. Maternal feeding practices may alter when fathers are present [39]. Thus, it is critical to distinguish between feeding roles of mothers and fathers in further research. In addition, measures used to assess parental non-responsive feeding practices and child eating behaviors were very different across studies. For example, most studies used validated questionnaires, while one used principal component analyses to explore the components of parental feeding practices and child eating behavior measures [30]. Thus, discrepancies in results may occur.

### 4.1. Limitations and Strengths

To the best of our knowledge, this review was among the first that comprehensively synthesized longitudinal data on the prospective relationships between parental non-responsive feeding practices and child eating behaviors. However, there are several limitations to this systematic review. First, although meta-analysis was conducted, the quantitative synthesis was mainly based on two or three studies because some studies did not report standard error or 95% CI, which precluded us from pooling all extracted data and from examining their bidirectional associations. Thus, the results from the meta-analysis should be interpreted with caution due to the limited number of studies included. Second, variability in the assessment of parental feeding practices (e.g., CFQ, IFSQ, and PFSQ) and child eating behaviors (e.g., CEBQ, CBCL, and FILAD) may make it difficult to compare findings; therefore, results should be interpreted with caution. Third, all included studies employed an array of self-reported questionnaires to assess our interest variables, which may be subject to recall bias. Furthermore, the included studies were mainly conducted in western countries such as Australia and the US. Findings from this review may not be extrapolated to other populations (e.g., Asian).

### 4.2. Implications

More longitudinal studies are needed to further examine the bidirectional relationships between parental non-responsive feeding practices and child eating behaviors. Such studies should be adequately powered and include a representative sample. Second, validated instruments should be used consistently across studies for future comparison between different studies. Third, more potential confounding factors (e.g., demographics, child temperament, and parental concern) should be taken into consideration. In addition, empirical studies should be derived from theoretical frameworks, such as Ecological System Theory [29], which contribute to a deeper understanding of the field.

## 5. Conclusions

There is mixed evidence for the bidirectional associations between parental non-responsive feeding practices and child eating behaviors. Only weak correlations were found between certain feeding practices and eating behaviors. Parents may be more likely to adopt restrictive feeding and food as a reward with children who have higher food responsiveness, whereas parents may be less likely to use restriction with children who have higher satiety responsiveness. Children may be more likely to have emotional eating if their parents use food as a reward more frequently. Future prospective, theory-driven studies using validated instruments and representative sampling while controlling for potential confounders are needed to provide more evidence. It is also important to understand the nature of their relationships, which may help to develop more personalized interventions to prevent childhood obesity. In addition, it is necessary to conduct relevant research in Asian and low-middle income countries, as all included studies were in western and high-income countries.

## Figures and Tables

**Figure 1 nutrients-14-01896-f001:**
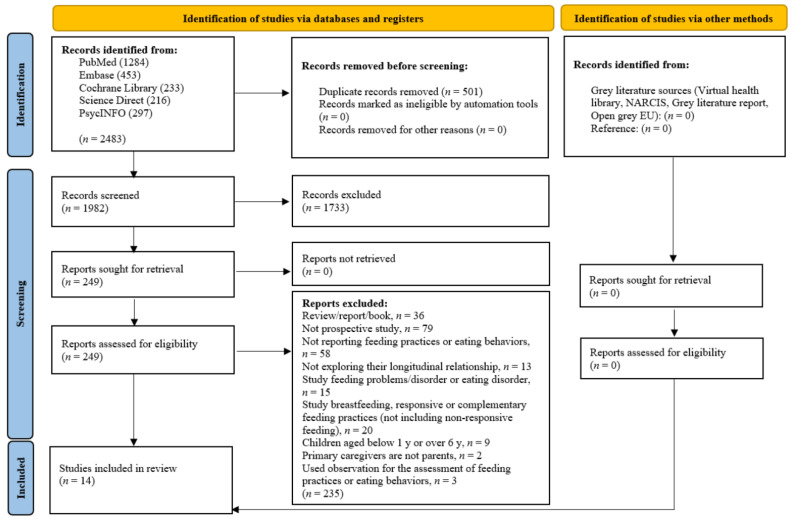
PRISMA flow diagram for screening and selection of articles.

**Figure 2 nutrients-14-01896-f002:**
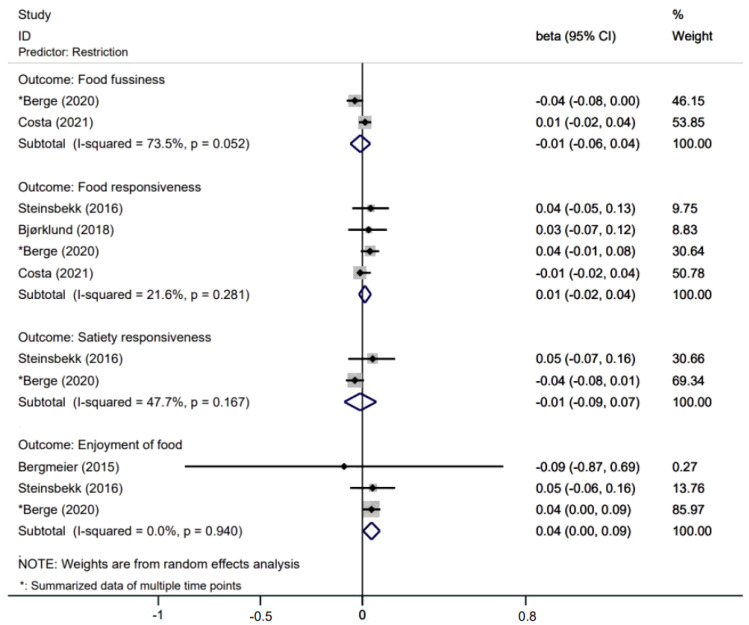
Effects of parental restriction on child eating behaviors.

**Figure 3 nutrients-14-01896-f003:**
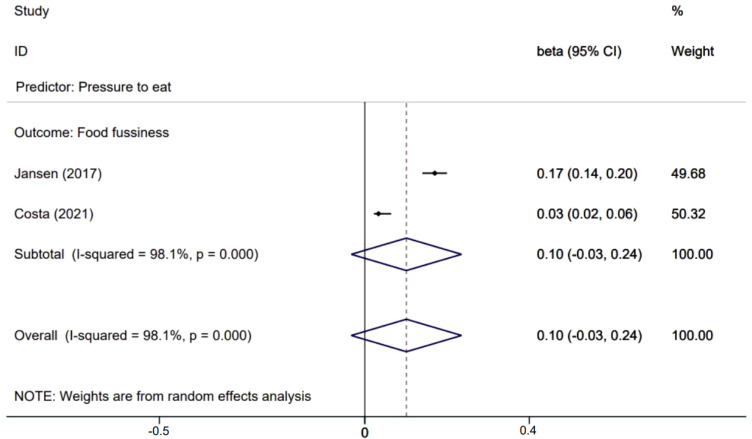
Effects of parental pressure to eat on child eating behaviors.

**Figure 4 nutrients-14-01896-f004:**
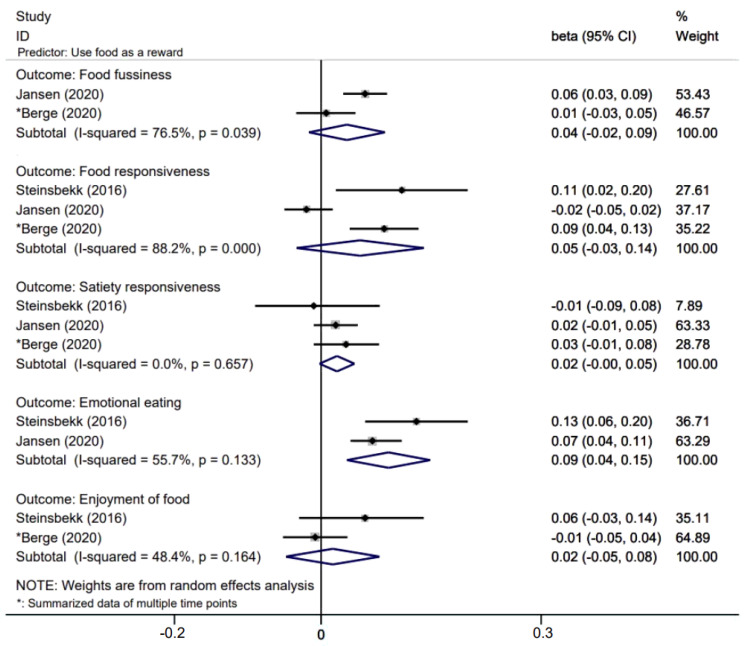
Effects of parental use of food as a reward on child eating behaviors.

**Figure 5 nutrients-14-01896-f005:**
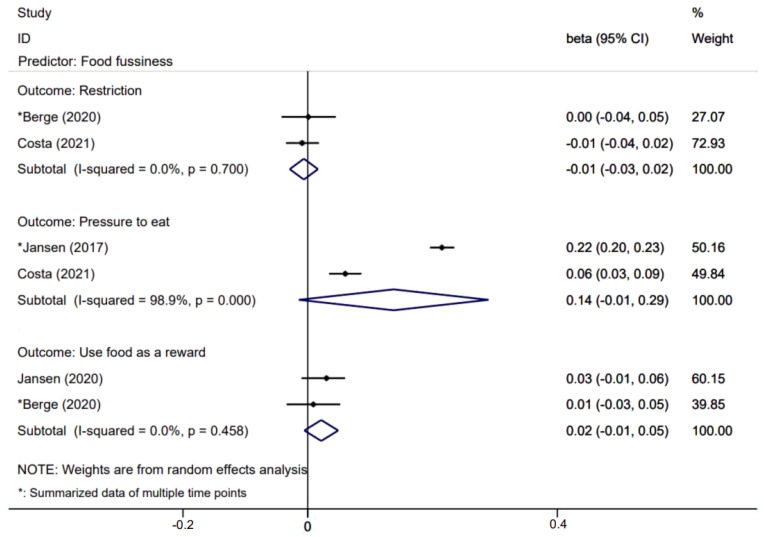
Effects of child food fussiness on parental non-feeding practices.

**Figure 6 nutrients-14-01896-f006:**
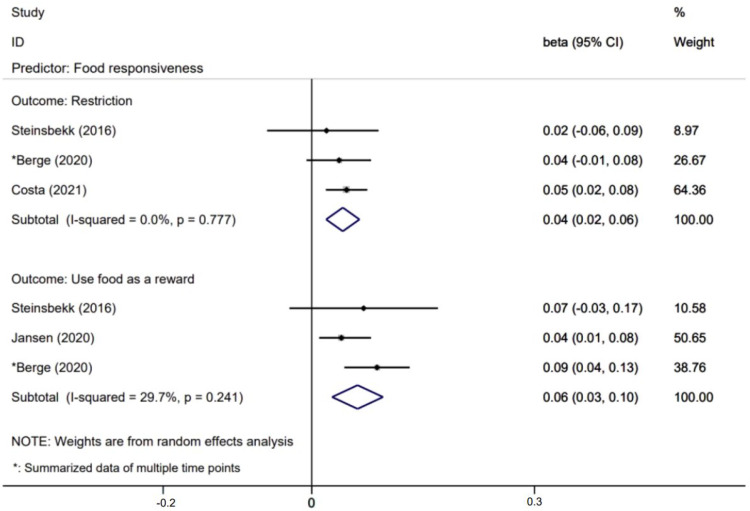
Effects of child food responsiveness on parental non-feeding practices.

**Figure 7 nutrients-14-01896-f007:**
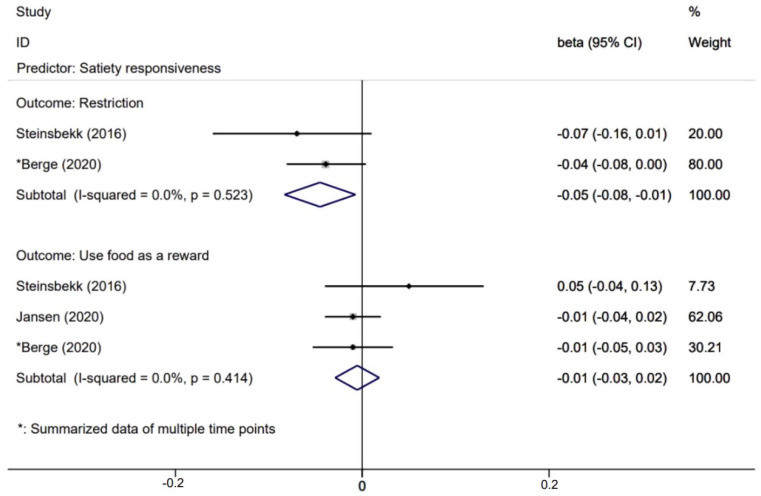
Effects of child satiety responsiveness on parental non-feeding practices.

**Figure 8 nutrients-14-01896-f008:**
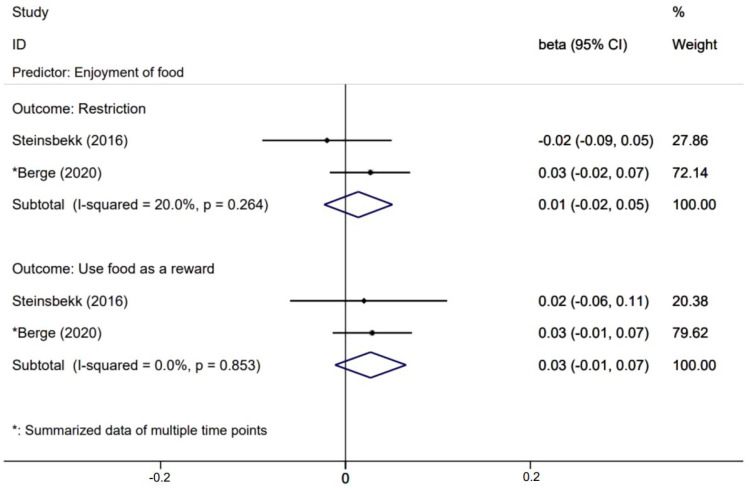
Effects of child enjoyment of food on parental non-feeding practices.

**Table 1 nutrients-14-01896-t001:** Characteristics of the studies (*n* = 14).

First Author, Year	Country	Frequency and Duration of Follow-Up	Caregivers	Age of Children at Recruitment	Sampling Method	Sample Size	Response Rate
Jansen et al., 2018 [14]	Australia	Three time points One for 1.7 years and one for 1.3 years	Mothers	2 years old.	Secondary data used a consecutive sample	207	59.83% (207/346)
Jansen et al., 2017 [31]	Netherlands	Four times for assessing children’s eating One for 1.5 years, one for one year and one for three years	Mothers	1.5 years old	Population-based sample	4845	66.41% (4845/7295)
Lumeng et al., 2018 [34]	US	Three time points Each for 6 months	Mothers	21 months old	Voluntary (response) sample	222	90.98% (222/244)
Mallan et al., 2018 [66]	Australia	Three time points One for 1.7 years and one for 1.3 years	Mothers	2 years old	Secondary data used a consecutive sample	207	59.83% (207/346)
Steinsbekk et al., 2016 [32]	Norway	Two time points Two years	Parents	6 years old	Voluntary (response) sample	623	78.17 (623/797)
Rodgers et al., 2013 [30]	Australia	Two time points One year	Mothers	1.5 years old	Voluntary (response) sample	222	68.70% (222/323)
Bergmeier et al., 2015 [40]	Australia	Two time points One year	Mothers	2 years old	Part of a longitudinal data which was 2-stage clustered sampling	72	91.14% (72/79)
Gregory et al., 2010 [52]	Australia	Two time points One year	Mothers	2 years old	Voluntary (response) sample	156	85.25% (156/183)
Bjørklund et al., 2018 [64]	Norway	Three time points Each for two years	Parents	6 years old	A representative community sample	702	88.08% (702/797)
Bergmeier et al., 2014 [67]	Australian	Two time points One year	Mothers	2 years old	Voluntary (response) sample	201	100.00% (201/201)
Zohar et al., 2020 [62]	US	Three time points Each for one year	Mothers	3.33 years old	Voluntary (response) sample	215	100.00% (215/215)
Jansen et al., 2020 [53]	Netherlands	Two time points Five years	Parents	4 years old	Population-based sample	3642	80.17% (3642/4543)
Berge et al., 2020 [63]	US	Four time points Each for 12 months	Parents	3.3 years old	Random sampling	534	100% (534/534)
Costa et al., 2021 [65]	Portugal	Two time points Three years	Mothers	4 years old	Population-based sample	3500	94.65% (3500/3698)

**Table 2 nutrients-14-01896-t002:** Key variables and related measurements (*n* = 14).

First Author, Year	Measures and Variables Related to Non-Responsive Feeding	Measures and Variables Related to Eating Behaviors	Covariates in the Final Model
Jansen, et al., 2018 [14]	FPSQ ① Overt restriction ② Reward for eating ③ Reward for behavior	CEBQ ① Satiety responsiveness ② Food responsiveness	Child BMI z-score at 14 months
Jansen et al., 2017 [31]	CFQ Pressure to eat	CBCL/CEBQ Fussy eating	Maternal ethnicity, education, psychopathology score and BMI, child gender, and breastfeeding duration
Lumeng et al., 2018 [34]	IFSQ Pressure to eat	CEBQ-T/BAMBI Picky eating	Did not report the covariates
Mallan et al., 2018 [66]	FPSQ ① Reward for Behavior ② Reward for Eating ③ Overt Restriction	CEBQ Food Fussiness	Child gender and maternal education did not substantively change any of the models The models did not include any covariates in the final model
Steinsbekk et al., 2016 [32]	PFSQ ① Instrumental feeding ② Control overeating	CEBQ ① Food responsiveness ② Enjoyment of food ③ Emotional overeating ④ Satiety responsiveness	Child BMI, parental BMI, and family socioeconomic status (SES) at age 6 years
Rodgers et al., 2013 [30]	① Instrumental feeding (PFSQ+CFPQ+CFQ) ② Emotional feeding (PFSQ+PFQ) ③ Control (PFSQ+COEQ) ④ Pushing to eat more (PFQ) ⑤ Fat restriction (CFQ) ⑥ Weight restriction (CFPQ)	DEBQ-P Emotional eating	T1 feeding practice and eating behaviors
Bergmeier et al., 2015 [40]	CFQ ① Restriction ② Pressure to eat	CEBQ ① Food fussiness ② Enjoyment of food	Maternal education, child BMI Z-score at T1, maternal control and concern about child weight, and child difficult temperament
Gregory et al., 2010 [52]	CFQ ① Pressure to eat ② Restriction	CEBQ ① Food responsiveness ② Food fussiness ③ Enjoyment of food	Maternal age, BMI and education, and child age; gender; T1 eating behavior; and T1 feeding practices
Bjørklund et al., 2018 [64]	① Control overeating (PFSQ) ② Instrumental feeding (PFSQ) ③ Restriction of food and avoidance of certain types of food (EDE-Q)	CEBQ Food responsiveness	Child and parent BMI
Bergmeier et al., 2014 [67]	CFQ ① Restriction ② Monitoring ③ Pressure to eat	CEBQ ① Food Fussiness ② Enjoyment of food	Maternal educational achievement, family income, and maternal BMI Child temperament, maternal Warmth and Control, and mother–child dysfunctional interaction T1 child eating behaviors and child BMI Z-score
Zohar et al., 2020 [62]	CFQ ① Monitoring ② Restriction ③ Pressure to eat	FILAD Picky eating	Birth order, child temperament, child executive function, and child sex
Jansen et al., 2020 [53]	CFQ Use food as a reward	CEBQ ① Food responsiveness ② Emotional overeating ③ Satiety responsiveness SFQ Fussy eating	Child BMI, sex, and ethnicity Parental BMI
Berge et al., 2020 [63]	PFSQ ① Emotional feeding ② Instrumental feeding ③ Control overeating	CEBQ ① Food responsiveness ② Satiety responsiveness ③ Food fussiness ④ Enjoyment of food	Child age, child sex, child BMI, child race, household maximum education attainment, household income, BMI of the primary adult respondent, and treatment group assignment.
Costa et al., 2021 [65]	CFQ ① Monitoring ② Restriction ③ Pressure to eat	CEBQ ① Food responsiveness ② Food fussiness	Child sex and BMI Z-score at 4 y of age and maternal education

Notes. IFSQ: Infant Feeding Styles Questionnaire; FPSQ: The Feeding Practices and Structure Questionnaire; CEBQ: Children’s Eating Behavior Questionnaire; CFQ: The Child Feeding Questionnaire; CBCL: Child Behavior Checklist; PFQ: Preschooler Feeding Questionnaire; COEQ: Control over Eating Questionnaire; CFPQ: Comprehensive Feeding Practices Questionnaire; PFSQ: The Parental Feeding Style Questionnaire; SFQ: The Stanford Feeding Questionnaire; FILAD: Foods I like and dislike; DEBQ-P: The parent version of the Dutch Eating Behavior Questionnaire; EDE-Q: Eating Disorder Examination Questionnaire; CEBQ-T: Children’s Eating Behavior Questionnaire-Toddler; BAMBI: The Brief Autism Mealtime Behavior Inventory; BMI: body mass index.

**Table 3 nutrients-14-01896-t003:** Summary of longitudinal effects from feeding practices to child eating behaviors (F→E).

	Food Fussiness	Food Responsiveness	Satiety Responsiveness	Emotional Eating	Enjoyment of Food	Total
Restriction	0/7 (0/10)	0/6 (0/9)	0/3 (0/6)	0/1 (0/1)	1/5 (1^−^/7)	1/10 (1^−^/33)
Pressure to eat	2/7 (2^+^/10)	1/2 (1^+^/2)	0/0 (0/0)	0/0 (0/0)	2/3 (2^−^/3)	4/7 (3^+^, 2^−^/15)
Use food as a reward	2/3 (2^+^/8)	3/5 (4^+^/10)	0/4 (0/9)	2/2 (2^+^/2)	0/2 (0/4)	5/6 (8^+^/33)
Emotional feeding	0/1 (0/3)	1/1 (2^+^/3)	1/1 (1^+^, 1^−^/3)	1/1 (1^+^/1)	0/1 (0/3)	2/2 (4^+^, 1^−^/13)
Total	4/10 (4^+^/31)	4/7 (7^+^/24)	1/4 (1^+^, 1^−^/18)	3/3 (3^+^/4)	2/5 (3^−^/17)	10/14 (15^+^, 4^−^/94)

Notes. A/B (C^+^, D^−^/E). A = number of articles with statistically significant associations; B = number of articles; C^+^ = number of positive associations; D^−^ = number of negative associations; E = total number of tested associations.

**Table 4 nutrients-14-01896-t004:** Summary of longitudinal effects from child eating behaviors to feeding practices (E→F).

	Food Fussiness	Food Responsiveness	Satiety Responsiveness	Emotional Eating	Enjoyment of Food	Total
Restriction	0/3 (0/6)	1/4 (1^+^/7)	1/3 (1^+^/6)	0/1 (0/1)	0/2 (0/4)	2/5 (2^+^/24)
Pressure to eat	2/3 (3^+^/7)	1/1 (1^−^/1)	0/0 (0/0)	0/0 (0/0)	0/0 (0/0)	2/3 (3^+^, 1^−^/8)
Use food as a reward	1/3 (1^+^/8)	2/4 (3^+^/9)	1/4 (1^−^/9)	1/2 (1^+^/2)	1/2 (1^+^/4)	3/6 (6^+^, 1^-^/32)
Emotional feeding	1/1 (1^−^/3)	0/1 (0/3)	0/1 (0/3)	1/1 (1^+^/1)	0/1 (0/3)	2/2 (1^+^, 1^−^/13)
Total	4/6 (4^+^, 1^−^/24)	3/5 (4^+^, 1^−^/20)	2/4 (1^+^, 1^−^/18)	2/3 (2^+^/4)	1/2 (1^+^/11)	7/9 (12^+^, 3^−^/77)

Notes. A/B (C^+^, D^−^/E). A = number of articles with statistically significant associations; B = number of articles; C^+^ = number of positive associations; D^−^ = number of negative associations; E = total number of tested associations.

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
