# Peer review of "Bidirectional Associations between Parental Non-Responsive Feeding Practices and Child Eating Behaviors: A Systematic Review and Meta-Analysis of Longitudinal Prospective Studies"

_nutrients, 2022, doi:10.3390/nu14091896_

Round 1

Reviewer 1 Report

This is a well-written paper. The method used for analysis is clear. The results and discussion focus primarily on definitive findings for rewarding with food and food responsiveness. It might be helpful, though perhaps not critical to the paper, to address their report of mixed findings related to other bi-directional characteristics that were weak or did not exist.

Edit text for errors in sentence structure or missing words.

Well-written literature review and methodology sections.

Reviewer 2 Report

In my opinion the manuscript entitled "Bidirectional associations between parental non-responsive feeding practices and child eating behaviors: A systematic review and meta-analysis of longitudinal prospective studies" presents an analysis of an interesting issue regarding the nutrition of children, but I have doubts whether it can be accepted for publication in the current version. It seems that the number of scientific studies that have been included in this meta-analysis is too small for the inference to be reliable, as Authors of the manuscript also write about. On the other hand, Authors performed the meta-analysis correctly and indicated the limitations of the study. My only objection concerns the figures that are hardly legible.
